# Photoautotrophs–Bacteria Co-Cultures: Advances, Challenges and Applications

**DOI:** 10.3390/ma14113027

**Published:** 2021-06-02

**Authors:** Viviana Scognamiglio, Maria Teresa Giardi, Daniele Zappi, Eleftherios Touloupakis, Amina Antonacci

**Affiliations:** 1Institute of Crystallography, National Research Council, Via Salaria Km 29.300, Monterotondo, 00015 Rome, Italy; viviana.scognamiglio@ic.cnr.it (V.S.); mt.giardi@biosensor.it (M.T.G.); daniele.zappi@ic.cnr.it (D.Z.); 2Biosensor S.r.l., Via Olmetti 44, 00060 Formello, Italy; 3Research Institute on Terrestrial Ecosystems, National Research Council, Via Madonna del Piano 10, 50019 Sesto Fiorentino, Italy; eleftherios.touloupakis@cnr.it

**Keywords:** microalgae, co-culture, bacteria, bio-molecules, biomass, sustainability, omics

## Abstract

Photosynthetic microorganisms are among the fundamental living organisms exploited for millennia in many industrial applications, including the food chain, thanks to their adaptable behavior and intrinsic proprieties. The great multipotency of these photoautotroph microorganisms has been described through their attitude to become biofarm for the production of value-added compounds to develop functional foods and personalized drugs. Furthermore, such biological systems demonstrated their potential for green energy production (e.g., biofuel and green nanomaterials). In particular, the exploitation of photoautotrophs represents a concrete biorefinery system toward sustainability, currently a highly sought-after concept at the industrial level and for the environmental protection. However, technical and economic issues have been highlighted in the literature, and in particular, challenges and limitations have been identified. In this context, a new perspective has been recently considered to offer solutions and advances for the biomanufacturing of photosynthetic materials: the co-culture of photoautotrophs and bacteria. The rational of this review is to describe the recently released information regarding this microbial consortium, analyzing the critical issues, the strengths and the next challenges to be faced for the intentions attainment.

## 1. Introduction

The benefits related to the photosynthetic microorganism’s cultivation (e.g., microalgae, cyanobacteria and photosynthetic bacteria) are well known owing to their ability to convert light, water and carbon dioxide into products with valuable contents in a sustainable way.

The ability of these photoautotrophs to manufacture high-value compounds (e.g., vitamins, pigments and polyunsaturated fatty acids), alternative energy sources and to perform natural processes for environment safeguard (e.g., biofuel production, CO_2_ mitigation and waste water treatment) founded a significant market demand from industries of food, feed, oil business, cosmetics and pharmaceutics [1,2,3,4,5]. In particular, the positive feedback related to the photoautotrophs’ use can be associated to their positive impact on the Industry 4.0 achievement. This strategic term was introduced in 2011 by the German government to indicate a technological perspective able to support sustainability challenges [6]. The expectations from the companies that embrace this new philosophy are many, including economic (as a result of a better energy use), environmental (for the reduction of scrap waste manufacturing and use of sustainable materials) and social ones (by achieving a safer and more comfortable workplace) [7]. In this context, microalgae and cyanobacteria exploitation received an increased interest for their attractive potential in the current scenario of energy savings and alternative food supplies. Many advantages derive from the use of these unicellular organisms, being capable of rapidly growing in a wide range of habitats as flexible systems. In addition, their strains can be easily edited by genetic approaches, and their cells have a shorter life cycle (1–4 days for microalgae and only 3 h for cyanobacteria) than the plants (90–180 days) [8,9,10]. This aspect is not to be underestimated, being that feedstocks with shorter life cycles are more sustainable in comparison with longer life ones [11].

However, the photoautotroph cultivations as long as they are in pilot systems and under controlled growth conditions have not shown particular worries; on the contrary, when industrial applications at large scales are reached, several concerns are brought up. In particular, two aspects heavily affect these scaled-up systems, i.e., the design of photobioreactors and the apparatus dedicated to the biomass harvesting. To this aim, evaluation in silico by mathematical simulations and dynamic modelling can be extremely useful for optimizing important parameters and for reducing the cost of expensive experiments. Process simulation is well established in chemical engineering, and its application in bio-engineering is increasing [12]. In this context, Apel and colleagues proposed a mathematical programming based on algorithms consisting of two loops: validation loop and virtual design and evaluation loop. The first one certifies that the models accurately describe reality. This step is strongly required to identify the parameters for the biology model, and it is repeated until the required accuracy is accomplished [12].

Moreover, another important issue related to the large-scale production is the culture contamination by other microorganisms (e.g., virus, bacteria, grazers and fungi) that can greatly reduce the biomass yield compromising the quality. In this regard, besides the contamination management strategies available into open ponds (e.g., use of adequate filters, selection of engineered strains, sterility conditions and antibiotics use), a new point of view is emerging on the concept contaminant organism, which is a new additional element capable of providing added value to the cultivation. In particular, the natural consortia have been a source of inspiration and offered the possibility to observe a new and complex reality able to combine productivity and high resistance to contaminations, because the ecological niche is already engaged. However, the final products of these natural consortia can be restricted due to the environment-elevated complexity. Recent efforts are focused on the design of stable, less complex and better controllable synthetic co-cultures to take advantage of microbial communities (e.g., distribution of the metabolic burden and ability to convert complex substrates) [13]. This synthetic ecological strategy is helpful for large-scale cultures, without sterile environments, and seems to be a promising approach to low-cost sustainable production of valuable compounds. Intriguing studies pointed out many strategies, such as engineering of chemical symbiosis, of quorum sensing and of ecological niches useful for this topic [14,15,16,17]. These parameters can be modulated purposefully to obtain a stable co-culture, in stress-free conditions and with the possibility to live together, making the most of the substrates partitioning as available resources [13]. In general, co-culture members can be identified according to the communication profile, which can be based on the analyses of metabolite content and protein assay, and/or taking inspiration from natural consortia. However, great challenges remain in the carrying out of multispecies cultures for industrial applications [18,19].

Several types of microalgae-based consortia have been reported in the literature, including microalgae–bacteria, algae–fungi, microalgae–protists and the multibacteria and multialgae symbiosis systems [20,21,22]. Microalgae–bacteria co-cultivation systems are extremely promising biotechnological tools, as many recent studies have revealed a positive effect of microalgae–bacteria symbiosis on microalgal growth [23,24,25]. The advantageous association of microalgae and bacteria has been attributed to various factors, e.g., (i) bacteria stimulate microalgal growth by producing growth-promoting substances as well as vitamins (cobalamin, thiamine and biotin) and cofactors and by reducing dissolved O_2_ concentration [26]; (ii) microalgae produce O_2_ through photosynthesis which can be utilized by bacteria, which in turn produce CO_2_ photosynthetically fixed by microalgae [23,27,28,29]; and (iii) microalgae secrete complex compounds which serve as a source of carbon and nitrogen for bacteria. As an example, in co-culturing *Chlorella vulgaris* and Pseudomonas, the symbiotic bacterium showed a growth-promoting effect on the green microalga [30].

The topic of synthetic microbial consortia is still in an early infancy, and several challenges related to the intercellular communication and the design of stable/manageable systems must be still faced. In this review, the state of the art concerning the peculiar bio-materials obtainable through the photoautotrophs–bacteria co-cultures is presented, highlighting the challenges and future directions to support the development of synthetic microbial consortia.

## 2. Microbial Consortium for Photosynthetic Biomanufacturing Materials

In the last 40 years, several important scientific discoveries have strongly contributed to the recent and promising possibility to use consortia of microorganisms for the synthesis of valuable biomolecules. In particular, starting from the first experiments of microbiology and selection of remarkable microorganisms, the knowledge evolved until sophisticated approaches (e.g., production of recombinant proteins and DNA editing) made possible the microbial co-cultivations for environmental and human purposes [31] (Figure 1).

In fact, a great potential of mixed crops is the possibility to obtain a wide range of photosynthetic materials, such as astaxanthin, β-carotene, omega-3 fatty acids, phycocyanin, exopolysaccharides and bio-polymers, as well as alternative energy sources (e.g., hydrogen, H_2_) and high amounts of biomass useful for many applications [19]. Moreover, recent studies reported that co-cultures, being designed for converting/producing to specific substrates/products, are able to ensure a higher yield than monocultures modeling growth parameters (e.g., high light, temperature, pH and salinity) or applying nutrient starvation [24]. This was demonstrated in the production of exopolysaccharides by co-culturing of *Agaricus blazei* and *Chlorella vulgaris*, or lipids by *Chlorella pyrenoidosa* and *Rhodospiridium toruloides* and *Chlorella vulgaris* and *Trichosporonoides spathulate* [33,34,35].

As mentioned above, the role of microalgae in the design of co-cultures is arousing much interest in the scientific community, and, to date, this strong potential has not yet been fully portrayed. In particular, the number of studies related to consortia and co-cultures in the wastewater treatment, anaerobic digestion and in biomanufacturing, where microalgae play important roles of co-culture partners, is greatly increasing (Figure 2). In this regard, case studies divided by bioproducts are reported below in order to obtain a snapshot of the current situation.

### 2.1. Manufacturing of Biomolecules

Biological molecules, such as carbohydrates, lipids and carotenoids, are synthesized by living organisms and are involved in essential cellular functions. Some of them have important roles in the marketing area, being pivotal in human health and diseases prevention, as well as being a sustainable resource for the environment (Figure 3).

A remarkable example is the integration of co-cultures in the enhancing of lipid production, positively impacting on yield and costs. In this context, Berthold and colleagues presented intriguing results of renewable fuel resources exploiting microalgae biomass, already known for their capability to produce several eco-fuels (e.g., diesel, hydrocarbons, biogas and ethanol) [36]. In this study, the authors avoid the common condition of nutrients depletion applied on monocultures to reach lipids accumulation, reason of inhibition in the biomass productivity, preferring a co-culture condition based on microalgae and bacteria isolated from 10 South Florida water samples [37]. In particular, effective results have been obtained with the algae *Characium* sp. and the heterotrophic bacterium, *Pseudomonas composti*, reaching a 50% increase in lipids content.

Another noteworthy research related to the effects of artificial consortia of microalgae and bacterial symbionts was presented by Xue and colleagues. In this case, *Chlorella vulgaris* and *Stenotrophomona smaltophilia* were analyzed in terms of biomass, growth rate and lipid production reaching an increase of 21.9%, 20.4% and 18%, respectively [38]. In particular, the co-culture condition affected not only the total content of lipids, but also their composition fostering the saturated fatty acids and oleic acids production. All these results highlight the great potential of microalgae–bacteria consortia in the quality of biodiesel. In addition, the results presented by Wei and colleagues supported the consortia exploitation, contributing to the demonstration that the quality and quantity of microalgal bio-oil can be improved and maximized with this new approach. In fact, through the co-culture of *Chlorella vulgaris* and *Mesorhizobium sangaii* (initial ratio of 40:1), the authors reported an increase in biomass, lipid content and productivity of algae (1.5, 2.2 and 3.3 times higher than those of the pure algae culture) [39].

A further contribution was presented with the co-culture of the green algae *Chlorella minutissima* with *Escherichi coli* under mixotrophic conditions [40]. In addition, in this case the attempt to replicate a “real” algae cultivation system revealed an interesting algae response to the presence of heterotrophic organism, suggesting a successful symbiotic relationship. This study documented significant increases in total lipid and starch productivity compared to axenic cultures, suggesting that bacterial contamination was not detrimental to the production of biofuel precursors but a possible added value. However, the experts in this field do not hide that further investigations are strongly required, especially to clarify the chemical/physical interactions that may occur between populations of different species, as well as to better cope the scaled-up systems. In the context of microbial renewable energies, the production of bio-H_2_, as an interesting alternative to gasoline, deserves to be mentioned, especially in the presence of co-cultivation.

Nowadays, several studies on H_2_ production by microalgae and bacteria co-cultures have been reported [41,42,43,44]. For instance, studies shown that Chlamydomonas–bacteria co-cultures can enhance H_2_ production [43,45,46]. In these co-cultures, dissolved oxygen in the culture medium is consumed by bacteria, thus producing an anaerobic environment that is suitable for hydrogenase, the enzyme responsible for H_2_ production, activation. Microalgae and bacteria co-culturing enhanced starch accumulation and maintained protein content. These factors are also important in order to improve H_2_ production. Fakimi et al. demonstrated the possibility of attaining synergetic H_2_ production in microalgae–bacteria consortia [47]. In these co-cultures, bacteria produce acetic acid, which is consumed by microalgae, and anaerobic conditions as a result of bacterial growth. H_2_ production in these microalgae–bacteria consortia resulted in 60% more H_2_ production than the sum of the respective monocultures [47].

A recent review of Fakimi et al. provided a comparison of previously published data about H_2_ production in Chlamydomonas–bacteria co-cultures and their respective algal monocultures [41,48]. Bacterial partners, such as Pseudomonas sp. and *Bradyrhizobium japonicum*, in cultures incubated in tris-acetate-phosphate devoid of sulphur medium, improved H_2_ production by 22-fold and 17-fold of the pure algal cultures, respectively [49].

Moreover, several studies on H_2_ production by photosynthetic bacteria and bacteria co-cultures have been reported [50,51,52]. Kao et al. showed that *Clostridium butyricum* and *Rhodopseudomonas palustris* co-cultures improved H_2_ production yield (2.16 mol H_2_/mol sucrose) compared to pure cultures (*Clostridium butyricum* monoculture 1.77 mol H_2_/mol sucrose, *Rhodopseudomonas palustris* monoculture 1.64 mol H_2_/mol sucrose) [50,51,52]. In the work of Hitit et al., the dark- and photo-fermentation processes were simultaneously carried out in a single-stage process by using co-cultures of *Clostridium butyricum* and *Rhodopseudomonas palustris* in order to enhance H_2_ production [53]. In this study, the *Clostridium butyricum* bacteria (dark fermentation) convert carbohydrates to H_2_ and produce volatile fatty acids as by-products that are used by the purple non-sulphur bacteria (photo-fermentation) to produce H_2_.

Unlike the production of lipids, few data are still available in the literature on the production of nutraceutical compounds, such as carotenes and xanthophylls, under photoautotrophic/bacterial co-culture conditions. In fact, the only results reported at this time describe other types of consortia (e.g., fungi/yeasts and algae/fungi), stating that this topic has not yet been fully focused [54,55]. However, some studies attest the accumulation of chlorophyll, ß-carotene, lutein and violaxanthin in microalgae co-cultivated with bacteria [56,57].

Another emergent sector within consortia of photoautotrophic organisms/bacteria concerns the production of exopolysaccharides (EPS), sugar-based molecules with promising traits for medical and pharmaceutical purposes, such as their capability to stimulate the immune system [58,59]. In this context, Angelis and colleagues investigated the EPS production in co-cultures of microalgae/cyanobacteria [60]. The reported results present an innovative technological process to increase the EPS production, both from algae and fungus, but with a prevalence of the latter, demonstrating a synergic effect and not a merely sum. Moreover, several studies report with increased interest the use and recovery of sludge, obtained from wastewater treatment, as recovery matrices and production of EPS, given the abundance of microbial species within these glomerular sludge [61,62,63].

The technology of mutualism and/or commensalism consortia paved the way to the fulfilment of novel platforms to produce bioplastics. In detail, polyhydroxyalkanoate molecules (PHA) are biodegradable and sustainable coproducts, particularly appealing for biorefinery markets, being useful for many applications currently covered by petroleum-based plastics [64,65]. Several microbial co-cultures of photosynthetic microorganisms and heterotrophic bacteria have been tested and evaluated for PHA bioplastic production [66,67], such as cyanobacteria/heterotrophic bacteria [68], photosynthetic bacteria/microalgae [69], photosynthetic bacteria/heterotrophic bacteria [70] and mixed purple non-sulphur bacteria [71]. Dinesh and colleagues used a combined dark- and photo-fermentation using co-cultures for H_2_ and poly-β-hydroxybutyrate (PHB) bioplastic production [72]. They obtained the maximal H_2_ and PHB production yields of 1.82 ± 0.01 mol H_2_/mol glucose and PHB 19.15 ± 0.25 g/L using 100% rice straw hydrolysate.

In more recent articles, the interactions between microalgae and prokaryotes in wastewater treatment are described [73,74,75,76,77,78,79,80]. Garcia et al. evaluated the performance of two photobioreactors operating with microalgae–bacteria and purple photosynthetic bacteria consortia during the treatment of diluted piggery wastewater [73]. Choi et al. studied the nutrient removal from artificial wastewater by the co-culture consortium of *Scenedesmus dimorphus* and nitrifier. The co-culture system showed enhancement in both nitrogen (N) and phosphorous (P) removal compared to each single culture [74]. da Silva Rodrigues et al. evaluated the bioremediation of sulfamethoxazole in wastewater treatment plant effluents using a tertiary treatment composed by microalgae–bacteria consortium. The microalgae–bacteria consortium used in this study demonstrated to be a promising alternative for bioremediation of sulfamethoxazole [79].

The new possibility of the use of microalgae–bacteria consortia for environmental and biotechnological purposes requires the use of efficient cultivation systems, such as photobioreactors. Many photobioreactor designs have been proposed and optimized for the cultivation of microalgae–bacteria consortia for the treatment of wastewater [77,81,82,83,84]. López-Serna and colleagues evaluated the removal of veterinary drugs from piggery wastewater in two open photobioreactors operated with a consortium of microalgae–bacteria and purple photosynthetic bacteria [85]. They found that a microalgae–bacteria consortium was more effective than purple photosynthetic bacteria in the removal of the detected veterinary drugs [85].

Algal–bacterial processes have been successfully tested for the treatment of domestic wastewater [81], centrates [86], vinasse [87] and piggery wastewater [88]. Wang and colleagues evaluated a Chlorella–Exiguo bacterium consortia cultured in a column photobioreactor for piggery wastewater purification [89]. They showed that the rates of contaminant removal by the Chlorella–bacteria consortium were higher than that of the pure Chlorella and bacterial monocultures. The results indicated that there is synergistic interactions of cell growth promotion and the exchange of carbon and oxygen between microalgae and bacteria [89]. Posadas et al. studied the photosynthetic biogas upgrading coupled with centrate treatment at pilot scale under outdoors conditions. In this work, the constant biomass productivity 15 g/m^2^/d and the minimization of effluent generation supported high nutrient recoveries in the harvested biomass (C = 66 ± 8%, N = 54 ± 18%, P ≈ 100%, S = 16 ± 3%) [86]. Serejo et al. confirmed the potential of algal–bacterial processes to support an efficient upgrading of biogas coupled to both wastewater treatment and the production of biomass [87].

### 2.2. Spent Biomass: An Efficient and Economically Viable Biorefinery Feedstock

The above-described potential of photosynthetic microorganisms/bacteria co-cultivations seems to be distinctly high in terms of yields and costs. Moreover, the residual production materials, e.g., biomass generated during the growth of the cultures, could appear as a hindrance to the sustainability of the entire production process because of the issues related to its disposal. Several intriguing studies pointed out the strong possibilities correlated to these biomaterials, as well as described sophisticated methods of harvesting and of reuse of organic matter (Figure 3). Noteworthy examples in the literature are related to the biomass recycling as feedstocks for biofuels manufacture [90,91], and animal feed preparation [92], whit additional benefits for bioeconomy.

This is the case of Hernández and colleagues study, which describes an integrated approach combining the production of second generation biofuels from the biomass of *Chlorella sorokiniana* with aerobic bacteria to produce biogas [94].

Recently, a huge development has been occurring in the treatment of municipal and industrial wastewater through the exploitation of biomass from algae/bacteria consortia (Figure 4). In fact, these co-cultures can play an important role in secondary and tertiary wastewaters treatment.

Furthermore, during wastewater treatments, symbiosis of microorganisms provides valuable benefits, such as cost-efficient aeration, removal of pollutants, sequestration of greenhouse gas emissions, as well as removing of pathogens (i.e., viruses), and flocs production, which means easier biomass managing [95]. For these reasons, promising results have been reported in the literature as well as innovative photobioreactor configuration with biomass recycling. An example can be the anoxic–aerobic algal–bacterial photobioreactor designed by Alcántara and colleagues to promote nitrogen removal via denitrification and the development of a rapidly settling algal–bacterial population [96]. This system efficiently removes organic carbon (86–90%), inorganic carbon (57–98%) and total nitrogen (68–79%) during synthetic wastewater treatment. In this context, Wicker and Bhatnagar described that slurry produced during biogas digestate treatment is a rich source of valuable nutrients not recommended to waste, due to the recent crisis of nutrient scarcity. However, chemical and recovery treatments are fiscally and energetically expensive. Hence, these authors proposed a compelling motion of nutrients remediation and resource recovery by a synergistic co-culture of eukaryotic and prokaryotic microorganisms to maximize the biomass production (1.99 g/L) [97].

Another fascinating study has been proposed by Goswami and colleagues, which highlighted the possibility to obtain a sustainable process for production of bio-crude oil via hydrothermal liquefaction of biomass generated by the co-cultivation of microalgae and bacteria coupled with wastewater remediation [98]. GC-MS and FTIR experiments on bio-crude oil was carried, and a great quantity of the hydrocarbon fraction was observed, as well as a high oil quality.

An alternate and promising reuse of biomasses derived from co-cultures of photoautotroph microorganisms and bacteria relies in its exploitations as biofertilizer in agriculture (Figure 5). The demand for biofertilizers is currently strongly growing, as they could be a valid alternative to toxic compounds widely used up to now on crops to ensure enhanced yields. In addition, biofertilizers play important functions in the decomposition of organic matter, thus ensuring greater nutrient availability for plants [99]. Finally, bio-fertilizers can increase the quantity and biodiversity of useful bacteria, such as plant growth-promoting rhizobacteria (e.g., Azotobacter, Bacillus, Burkholderia, Pantoea, Pseudomonas, Serratia and Streptomyces) [100,101].

The positive effects of three bacteria (*Providencia* sp., *Brevundimonas* sp. and *Ochrobacterium* sp.) and three cyanobacterial strains (*Anabaena* sp., *Calothrix* sp. and *Anabaena* sp.) were evaluated in a pot experiment with rice variety (Pusa-1460) by Prasanna and colleagues. In this study, the authors observed a highest yield enhancement of 19.02%, a significant enhancement in nitrogen fixing potential, and an improved soil health by N saving of 40–80 kg/ha, in particular with *Anabaena* sp. and *Ochrobacterium* sp. [103].

The combined bio-stimulant properties of algae and bacteria has been tested also by Kopta and colleagues on lettuce plant. The results demonstrated that the bacterial–algal preparation significantly affected the plant weight of both romaine and leaf lettuce in the spring and summer seasons [104]. The highest increase for romaine lettuce has been obtained in the spring crop (18.9%) and for leaf lettuce in the summer crop (22.7%). Moreover, the data pointed out that the photosynthetic compounds produced by algae, such as carotenoids, could improve the plant crops in terms of quality and quantity, and furnish support in stress conditions as can be present during the summer season. Analogues results, related to growth promotion, productivity and quality of the plants, were reported also for common bean, maize and onion by administering on crops a preparation of algae and bacteria [105,106,107]. For all these reasons, it is reasonable to suppose that co-cultures systems can be more effective in enhancing microbial diversity in the soil, resistance to plant diseases and productivity of vegetable crops. However, further investigations concerning the molecular mechanisms underlying the effect of microalgae–bacteria consortia on plants growth, and their disease suppression, are essential in light of to their safer and wider use in agriculture.

## 3. Improve the Consortia by Omics Approaches

Synthetic biology and metabolic engineering made great strides in constructing and optimizing metabolic pathways in model microbes [32,108]. These methodologies could be crucial for the exploitation of photoautotrophic/bacterial co-cultures, especially if the aim is to achieve a deeper knowledge of the molecular mechanisms underlying a consortium and a refinement of the features that could ensure a more stable and productive co-culture. Several experimental methods can contribute to clarifying physiological characteristics within the consortia, such as microscopy, cell sorting, mass spectrometry, molecular techniques and genetic engineering. Recently, the latter played an important role in the current challenge related to the implementation of cell–cell communication [109]. Although many studies are available in the literature concerning algal–bacterial interactions, notions regarding the chemical-ecological role and actors of this relationship remains limited, particularly with respect to quorum sensing, which is a system of stimuli and responses correlated to population density [110].

In general, omics approaches can offer important details on the relation between environmental factors and genes/transcriptomes/metabolites and protein expression, as well as in the gene regulation of consortia (Figure 6) [62]. In detail, genomic techniques can provide data on rapid and accurate species composition of the consortia, identifying genetic variabilities and comparison among the species [111]. However, these approaches require specific knowledge and genome libraries, and they can be inadequate for the isolation of low abundant species in consortia. Instead, the transcriptomics tools can be useful in the consortia study because they can provide data on specific transcripts regulation and transcriptional comparison between patterns of consortia and axenic cultures [112]. However, these techniques require long and complex sample preparations.

Concerning the proteomic approach, high-throughput analysis of proteome can be obtained, intriguing connections between genetic and biochemical information highlighted, regulatory mechanisms in consortium identified and enzymatic studies performed [113]. Nevertheless, data processing and analysis are very difficult and complex, for this reason highly skilled personnel are strongly required. Recently, the analysis of the total “secretoma” of a microbial community has also been very successful as it provides a clear summary of the proteins involved and their dynamics outside microbial cells. Indeed, the metasecretomics approach allows for a complete identification of the total surface-bound proteins and secretions in the consortia [114,115].

Finally, through reliable and reproducible metabolomics techniques, qualitative and quantitative data can be achieved on consortium-produced metabolites [116]. They are very sensitive techniques, sample preparation is easier than the techniques described above, but they require the use of hazardous solvents for the extraction of metabolites.

The potential of genetic engineering on algae–cyanobacteria/bacteria consortia and its benefits for industry begins to be demonstrated from the scientific literature even if few studies have been done actually. A noteworthy example in this context is certainly the research described by Ortiz-Marquez and colleagues, who edited the genome of *Azotobacter vinelandii* bacteria, obtaining higher capability of fixing nitrogen and ammonium release in the medium in comparison with control. This approach determined a growth stimulation on oil-producing microalgae and lipids accumulation [117].

Another interesting study has been recently provided by Ducan and collaborators, who described an improved synthetic photoautotroph/chemoheterotroph consortium with *Synechococcus elongatus* and the bacterium *Halomonas boliviensis*. This successful microbial blend is derived from the sucrose secreted by cyanobacteria in support of the bacterium, a natural producer of the bioplastic precursor, achieving a productivity level of 28.3 mg PHB L^−1^ d^−1^. Moreover, the trial carried out on this consortium pointed out a high productivity of at least 5 months and resistance to the attack of invasive microbial species without the use of antibiotics or chemicals [118].

## 4. Conclusions and Future Perspective

The exploitation of photoautotrophic organisms and bacteria co-cultures showed many advantages when compared to monocultures. In fact, such co-cultures are effective farms for the production of different biomaterials with strong interest in human health, as well as source of alternative and sustainable energies. This successful combination, inspired by nature, is based on several fruitful interactions between the cells of photoautotrophs and bacteria, based on exchanges of vitamins, oxygen and carbon dioxide. In addition, algae/bacteria consortia ensure the minimization of contamination, a serious issue in axenic cultures, and it can lead to greater resistance and stability in the cell population. These are important qualities during the scale-up of the cultures [120], and, from an industrial point of view [121,122,123], these are synonyms of higher yields and economic savings, although much can be done from the engineering side (e.g., design of bioreactors) and in the management of biomasses (e.g., cells harvesting).

Moreover, insights on transcriptomics, metagenomics and metabolomics are currently strongly required to reach a better understanding of microbial interactions, addressing the use of available substrates and increasing the productivity. Furthermore, a genome-editing approach could be the driving force to promote in the next future a sustainable production of molecules not conventionally synthesized by these microorganisms.

The analysis of the current scenario on microbial consortia highlights another important issue to face, which is the development and consolidation of computational and mathematical assistances for co-cultures realization. In fact, this aspect, which strongly affects the total costs and the time required to obtain large volume systems, is especially essential for the achievement of the real market.

Another desirable tool for achieving large-volume co-cultures, as suggested by Padmaperuma and colleagues, could be the realization of an open-access database collecting relevant metadata about tested consortia and their outcomes (both positive and negative), description of the selected strains, growth dynamics, biomolecules released and data related to the conditions in bioreactors if tested [24]. Undeniably, this would be a great support for the academic research and for the technology transfer from bench-scale to industrial applications.

## Figures and Tables

**Figure 1 materials-14-03027-f001:**
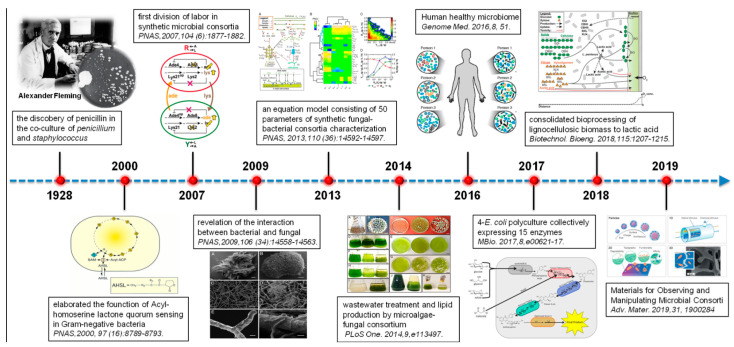
A schematic timeline representing the main milestones results related to the microbial consortia statement. Reprinted with permission from Qian et al. (2020). Biotechnological potential and applications of microbial consortia. Biotechnology Advances [32]. Copyright (2020) Elsevier.

**Figure 2 materials-14-03027-f002:**
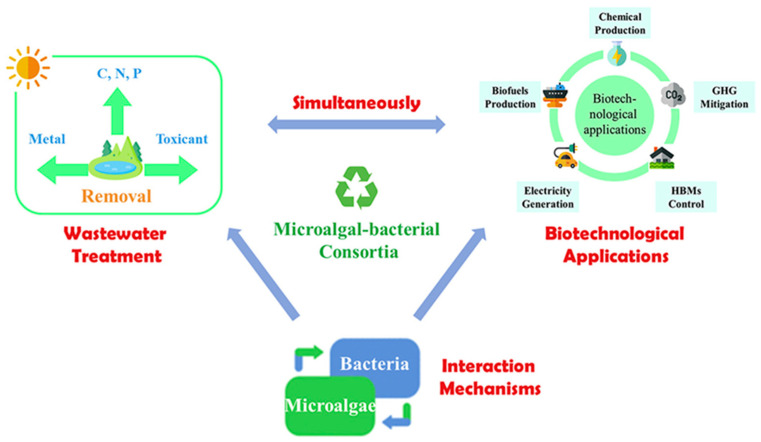
Microalgal–bacterial consortium and its biotechnological applications. Reprinted with permission from Zhang et al. (2020). Microalgal–bacterial consortia: From interspecies interactions to biotechnological applications. Renewable and Sustainable Energy Reviews [18]. Copyright (2020) Elsevier.

**Figure 3 materials-14-03027-f003:**
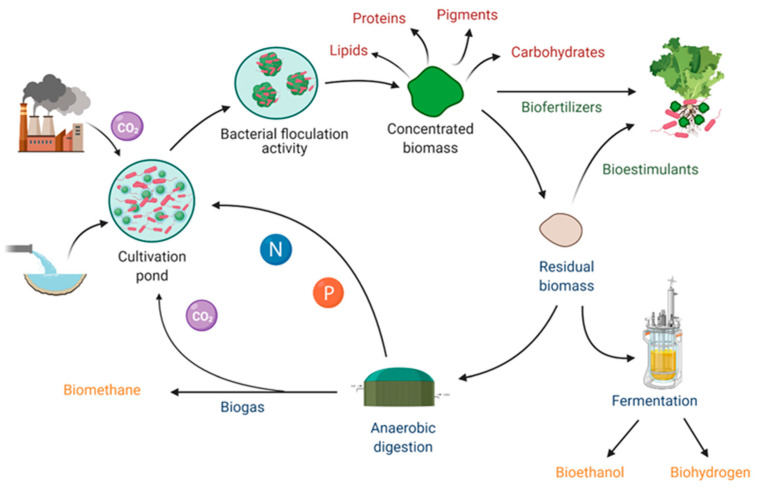
Schematic representation of the possible routes intended for microalgae–bacteria co-cultures in the biorefinery scenario. With the scale-up cultivation of these algae/bacteria consortia, there are many possible applications, e.g., the production of high-value-added compounds, reduction of CO_2_ emissions, and by the reuse of biomass the production of bio-fertilizers, water bio-sanitation, production of alternative and sustainable energies. Reprinted with permission from González-González et al. (2021). Toward the Enhancement of Microalgal Metabolite Production through Microalgae–Bacteria Consortia. Biology [93]. Copyright (2021) MDPI.

**Figure 4 materials-14-03027-f004:**
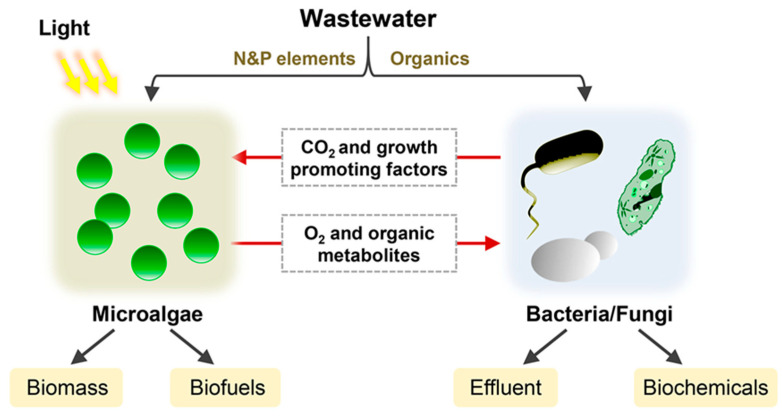
The synergistic interaction of microalgae and other microbes (such as bacteria) for wastewaters treatments. The consortium cells by metabolite exchange (organic compounds, O_2_, CO_2_ and growth-promoting factors) play important role in the secondary and tertiary treatment of these waters; moreover, the harvested biomass can further be used as feedstock for biofuel or biochemical production. Reprinted with permission from Qian et al. (2020). Biotechnological potential and applications of microbial consortia. Biotechnology advances [32]. Copyright (2020) Elsevier.

**Figure 5 materials-14-03027-f005:**
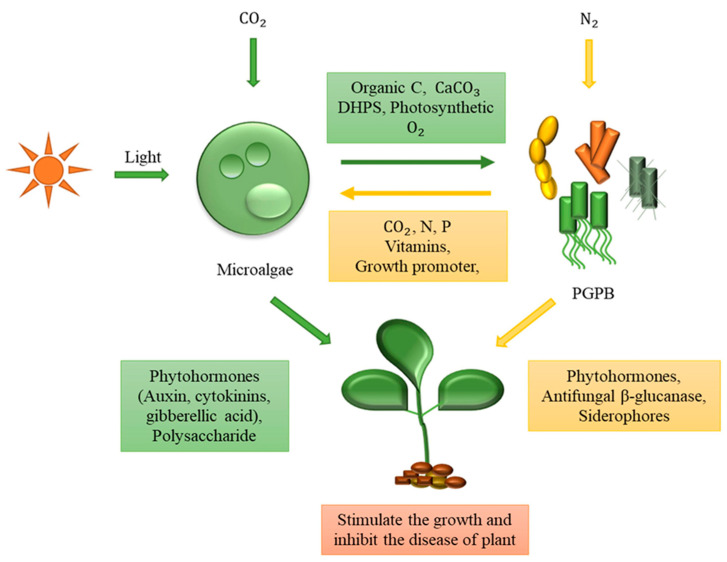
Consortia microalgae based can stimulate the plant growth-promoting bacteria (PGPB) allowing a sustainable agriculture. In detail, the microalgae cells and PGPB establish a beneficial exchange of nutrient molecules, and together enhance plant growth by producing phytohormones and other growth stimulants. Furthermore, this consortium can prevent the occurrence of certain plant diseases with particular suppressive mechanisms. Reprinted with permission from Kang et al. (2021). Potential of Algae–Bacteria Synergistic Effects on Vegetable Production. Frontiers in Plant Science [102]. Copyright (2021) Frontiers.

**Figure 6 materials-14-03027-f006:**
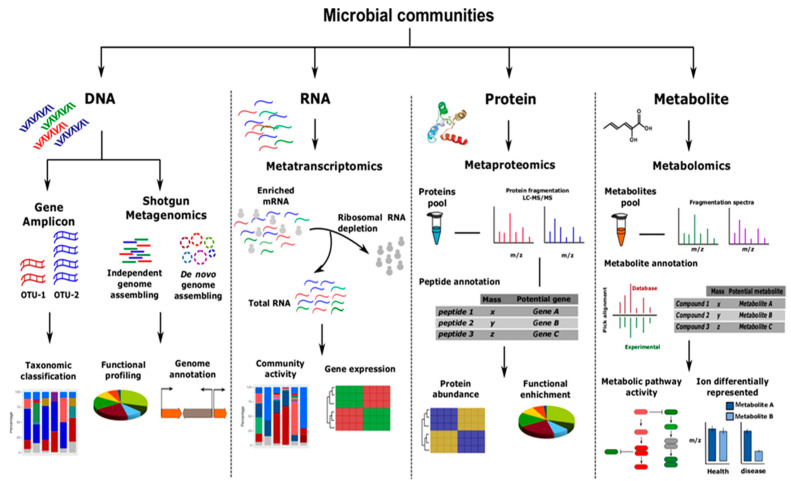
Omics techniques for the construction of an efficient consortium. Different experimental and computational methods can be applied on co-cultured cells to reach a deep knowledge and characterization of the microbial communities. Reprinted with permission from Zuñiga et al. (2017). Elucidation of complexity and prediction of interactions in microbial communities. Microbial biotechnology [119]. Copyright (2017), Wiley.

## Data Availability

The study did not report any data.

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
