# Peer review of "Photoautotrophs–Bacteria Co-Cultures: Advances, Challenges and Applications"

_materials, 2021, doi:10.3390/ma14113027_

Round 1
Reviewer 1 Report
This manuscript reviewed the photoautotrophs-bacteria co-cultures. The review is to describe the recently released information regarding this microbial consortium, analysing the critical issues, the strengths, and the next challenges to be faced for the intentions attainment. Challenges and limitations have also been identified. My recommendation is Minor revision. However, there are some issues of the manuscript need to be attention. Authors may see the comments as the follow:
- It is better to move the Figure 1 from introduction section. You may include it into other section.
- Some recent references form material about this topic are recommend to be cited in the manuscript.
Author Response
Point 1: It is better to move the Figure 1 from introduction section. You may include it into other section.

Response 1: We would thank the reviewer for the suggestions that we deeply followed. Figure 1 has been moved in the paragraph “2. Microbial consortium for photosynthetic bio-manufacturing materials” as kindly suggested.
Point 2: Some recent references form material about this topic are recommend to be cited in the manuscript.
Response 2: We would like to thank the reviewer for the precious comments. As kindly suggested, we added some more recent references from 2020 and 2021 highlighting the importance of photoautotrophs organisms-bacteria cultures in improved lab scale and industrial applications.
Reviewer 2 Report
The work “Photoautotrophs-bacteria co-cultures: advances, challenges and applications “ is very timely review on the biomanufacturing, provides an overview of the field while focusing on more recent works. In my opinion this review deserves to be published. I have minor suggestions in order to possibly improve the work presented.
Doubling time of fast-growing cyanobacteria can be as short as 2-3 hours.
Although the authors mention metabolomics, proteomics approaches, I suggest to emphasise ‘metasecretomics’-based approach.
A coupled engineered co-cultures can also be mentioned, e.g. engineered photosynthetic cyanobacteria produce and secrete sucrose which is consumed by E.coli to synthetize targeted chemicals (at least D. Ducat lab has published similar work).
Author Response
Point 1: Doubling time of fast-growing cyanobacteria can be as short as 2-3 hours.
Response 1: We would like to thank the reviewer for this important clarification regarding the doubling time of cyanobacteria, which we have inserted into the text of the manuscript (lines 55 and 56).
Point 2: Although the authors mention metabolomics, proteomics approaches, I suggest to emphasise ‘metasecretomics’-based approach.
Response 2: We agree with the reviewer’s idea, and we gladly accept this suggestion. For this reason we included in the paragraph “3. Improve the consortia by omics approaches”, the recent and promising analysis of metasecretomic (lines 420-424).
Point 3: A coupled engineered co-cultures can also be mentioned, e.g. engineered photosynthetic cyanobacteria produce and secrete sucrose which is consumed by E.coli to synthetize targeted chemicals (at least D. Ducat lab has published similar work).
Response 3: The example suggested by the reviewer to improve the engineering consortia section has been inserted in the text (lines 436-443).
Reviewer 3 Report
The axenic cultivation of microalgae in large scale bioreactors, mainly in heterotrophic conditions, is really a huge biotechnological problem. The possibility of using the forementoned problem as a profitable solution is a very clever idea.
The symbiotic behavior of microorganisms could be a good manufacturing process in future, since it can lead to higher efficiencies of valuable products.
In nature, a well-established symbiotic system is lichens (fungi with green algae or cyanobacteria). The mycobiont part consumes the oxygen, that deactivates the hydrogenase of the green algae (photobiont), and as a result large quantities of hydrogen are produced. You could also include lichens, as a very successful symbiotic system in nature, in your review.
The authors should increase the font size in Figures 1 (the wastewater treatment box and the scheme in biotechnological applications),2 (text in the all boxes),3 (only the words biomethane, bioethanol and biohydrogen) and especially 6 because it was very difficult to read them.
Line 189. The word However has two H accidentally.
Author Response
Point 1: In nature, a well-established symbiotic system is lichens (fungi with green algae or cyanobacteria). The mycobiont part consumes the oxygen, that deactivates the hydrogenase of the green algae (photobiont), and as a result large quantities of hydrogen are produced. You could also include lichens, as a very successful symbiotic system in nature, in your review.
Response 1: We would like to thank the reviewer for the precious suggestion regarding lichens; however, i) being the present review focused on photoautotrophs-bacteria co-cultures (e.g. microalgae, cyanobacteria and photosynthetic bacteria), and ii) being lichens a symbiotic association between fungi and algae/cyanobacteria, we feel that this latter argument deserves a separate review to be implemented and thus we not included it in the manuscript.
Point 2: The authors should increase the font size in Figures 1 (the wastewater treatment box and the scheme in biotechnological applications),2 (text in the all boxes),3 (only the words biomethane, bioethanol and biohydrogen) and especially 6 because it was very difficult to read them.
Response 2: All the figures we inserted in the manuscript have been extrapolated from others articles with permissions from the journals. For this reason, we cannot modify them but surely we will ask to the editors to publish them with the right definition and dimensions.